# CTFFIND5 provides improved insight into quality, tilt, and thickness of TEM samples

**Johannes Elferich[1,2]\*, Lingli Kong[1], Ximena Zottig[1,2], Nikolaus Grigorieff[1,2]\***

[1]RNA Therapeutics Institute, University of Massachusetts Chan Medical School, Worcester, United States; [2]Howard Hughes Medical Institute, University of Massachusetts Chan Medical School, Worcester, United States

## eLife Assessment

This **valuable** work presents the latest version of CTFFIND, which is the most popular software for determination of the contrast transfer function (CTF) in cryo-electron microscopy. CTFFIND5 estimates and considers acquisition geometry and sample thickness, which leads to improved CTF determination. The paper describes **compelling** evidence that CTFFIND5 finds better CTF parameters than previous methods, in particular for tilted samples (e.g. for cryo-electron tomography) or where thickness is an issue (e.g. cellular samples, or electron microscopy at low voltages).

**\*For correspondence:**
Johannes.Elferich@umassmed.
edu (JE);
niko@grigorieff.org (NG)

**Abstract** Images taken by transmission electron microscopes are usually affected by lens aberrations and image defocus, among other factors. These distortions can be modeled in reciprocal space using the contrast transfer function (CTF). Accurate estimation and correction of the CTF is essential for restoring the high-resolution signal in cryogenic electron microscopy (cryoEM). Previously, we described the implementation of algorithms for this task in the *cis*TEM software package (Grant et al., 2018). Here we show that taking sample characteristics, such as thickness and tilt, into account can improve CTF estimation. This is particularly important when imaging cellular samples, where measurement of sample thickness and geometry derived from accurate modeling of the Thon ring pattern helps judging the quality of the sample. This improved CTF estimation has been implemented in CTFFIND5, a new version of the *cis*TEM program CTFFIND. We evaluated the accuracy of these estimates using images of tilted aquaporin crystals and eukaryotic cells thinned by focused ion beam milling. We estimate that with micrographs of sufficient quality CTFFIND5 can measure sample tilt with an accuracy of 3° and sample thickness with an accuracy of 5 nm.

## Introduction

Transmission electron microscopy of biological specimens at cryogenic temperatures (cryoEM) has become a widely used method to image biomolecules at high resolution, both in solution and within the cell. To retrieve the high-resolution signal, the cryoEM images have to be corrected for the contrast transfer function (CTF) of the microscope. Common parameters used to describe the CTF include an astigmatic defocus, the spherical aberration of the objective lens, and if appropriate, a phase shift introduced by a phase plate. The defocus and phase shift parameters are commonly estimated by fitting the Thon ring pattern (**Thon, 1971**) in the power spectrum of micrographs to a modeled power spectrum. The program CTFFIND4 (**Rohou and Grigorieff, 2015**) has been developed for this task and the model and conventions to describe the CTF are widely adopted in the field.

A limitation of CTFFIND4 is that it considers the whole imaged sample to be at the same objective defocus, which is a reasonable assumption for flat and thin samples, as is common in single-particle cryoEM. However, the increased thickness of cryoEM samples of cells may introduce

additional modulations in the Thon ring pattern (*Tichelaar et al., 2020*) that can lead to errors in the CTF modeling when not accounted for. Furthermore, samples of cells are often tilted with respect to the optical axis of the microscope, either unintentionally due to thinning methods such as cryogenic focused ion beam (FIB) milling, or intentionally during electron cryo-tomography imaging. In both cases the effects are strongest at high resolution, where the Thon rings are more tightly spaced.

Here, we describe new features of CTFFIND5 that can fit the modulations of the Thon ring patterns and determine sample thickness and tilt using an extended CTF model with additional parameters. This not only increases the fidelity of the fit, as Thon rings at higher resolution can now be fitted reliably, but also gives valuable insight into the geometry of the sample that can aid the experimentalist.

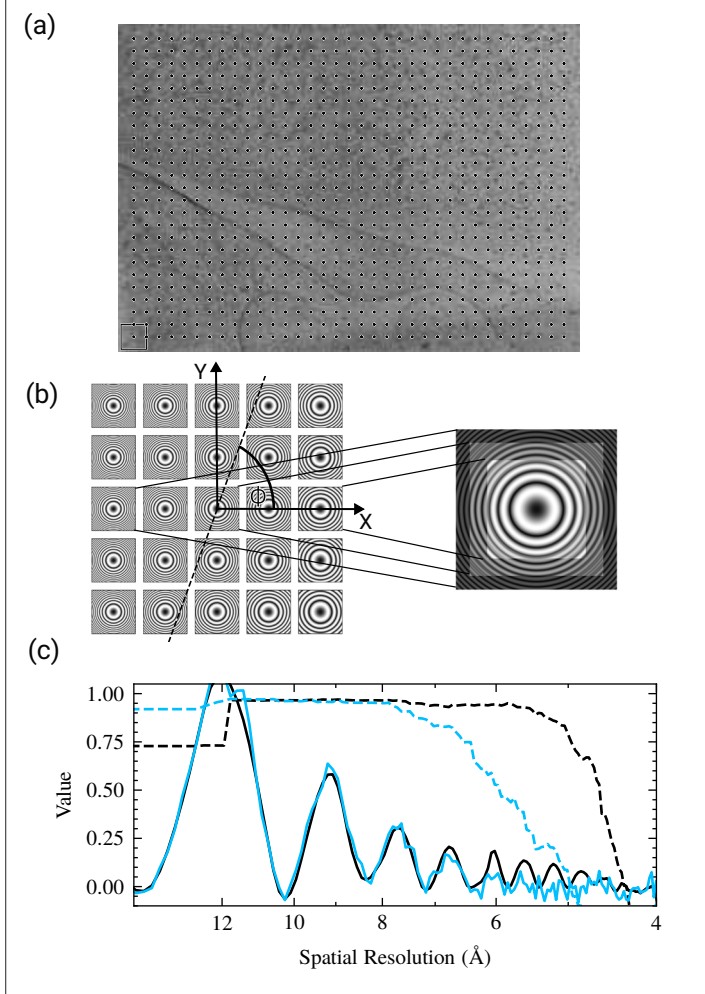

**Figure 1.** Tilt estimation and correction in CTFFIND5. (**a**) Power spectra are calculated in 128×128 pixel patches as indicated on a representative micrograph. The dots represent the locations of the patches and the box indicates patch size. (**b**) A model of the expected power spectrum in each patch given an average defocus $\Delta f$, tilt angle $\theta$, and tilt axis $\phi$ is compared to the actual power spectra of tiles. After an optimal set of $\theta$ and $\phi$ has been found a corrected power spectrum is calculated by summing the tile power spectra, scaled to correct for the defocus difference. Power spectra shown are an exaggerated example. The convention of $\phi$ as a counterclockwise rotation from the x-axis is indicated. (**c**) Comparison of the original power spectrum (EPA, solid line, blue) to the tilt-corrected power spectrum (solid line, black). The tilt-corrected power spectrum exhibits clear peaks at higher spatial resolution than the uncorrected power spectrum, as evident by the 'goodness-of-fit' scores (dashed lines). The estimated contrast transfer function (CTF) parameters are $\Delta f_1 = 10603\text{Å}, \Delta f_2 = 10193\text{Å}, \alpha = 85.9°$ for the uncorrected power spectrum and $\Delta f_1 = 10492\text{Å}, \Delta f_2 = 10342\text{Å}, \alpha = 81.2°, \theta = 12.3°, \phi = 261.6°$ for the tilt-corrected power spectrum. The fit resolution is 5.9 Å for the uncorrected power spectrum (dashed line, blue) and 4.6 Å for the tilt-corrected spectrum (dashed line, black).

## Methods

### Tilt estimation algorithm

Tilt estimation in CTFFIND5 follows a strategy that is similar to the implementation in CTFTILT (*Mindell and Grigorieff, 2003*). The tilt axis direction $\phi$ and tilt angle $\theta$ are determined by fitting Thon ring patterns locally, calculated from 128×128 pixel tiles that form a regular grid covering the micrograph (*Figure 1a*). In this model, $\phi$ has a positive value ranging from 0° to 360° and describes the angle of the tilt axis to the x-axis of the micrograph in the counterclockwise direction. The tilt angle $\theta$ has positive values ranging from 0° to 90° and describes the rotation of the sample around the tilt axis in a counterclockwise direction. It is assumed that the defocus variation across the sample can be described by a tilted plane. Fits are evaluated using correlation coefficients between modeled CTFs and Thon ring patterns. Initially, the micrograph pixel size is adjusted (binned) by Fourier cropping to match the resolution limit of the fit set by the user and the micrograph is cropped to be square in order to speed up computation. A power spectrum is calculated from this binned and cropped image, a smooth background is calculated using a box convolution (*Mindell and Grigorieff, 2003*) and subtracted, the power spectrum is further binned to the tile size (128×128 pixels), and the fit of the tilted Thon ring patterns across the micrograph is initialized by fitting this highly binned power spectrum with a non-astigmatic CTF. This fit is then refined using a two-dimensional CTF with astigmatism. Rough values for the tilt axis and angle are then determined in a systematic search in 10° and 5°

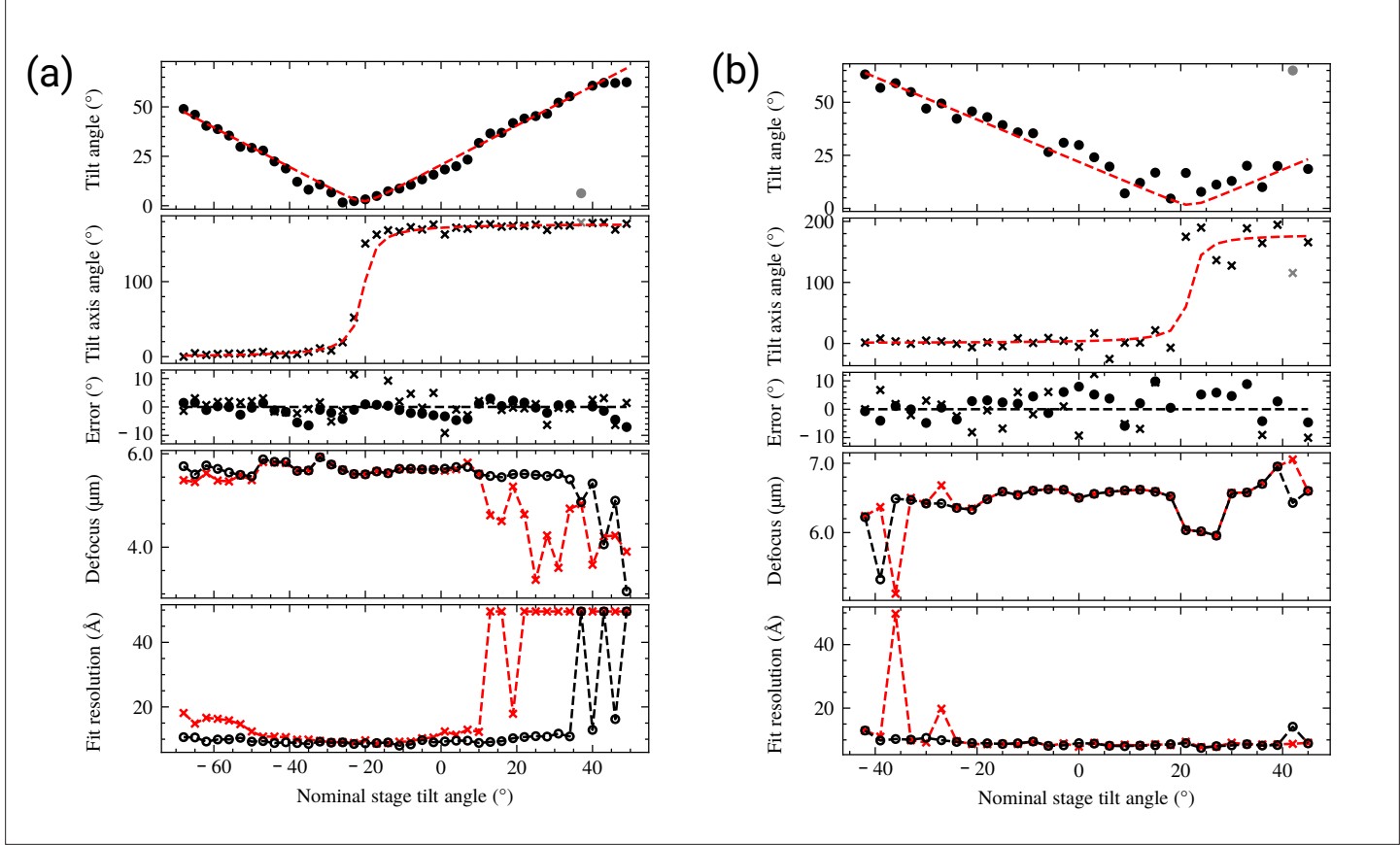

**Figure 2.** Validation of tilt estimation using tilt series data and comparison of defocus estimation using CTFFIND4 and CTFFIND5. (**a**) Estimated tilt angle and axis of 40 micrographs of a tilt series taken on a focused ion beam (FIB)-milled biological specimen. For each image the tilt angle (dots, top plot) and tilt axis direction (crosses, second plot) are plotted as a function of the nominal stage angle. The data were fitted to a model of the specimen tilt and constant stage tilt axis before tilting the stage (red dashed line in first and second plot). The estimated stage tilt axis has an angle of 178.2° and the estimated specimen pre-tilt is 20.6° with a tilt axis of 171.8°, which is consistent with the FIB-milling angle of 20° and manual alignment of the milling direction to the goniometer tilt axis. The third plot shows the fit residuals for tilt angle and axis are plotted. The fourth and fifth plots show a plot of the estimated defocus value and fit resolution for each tilt image, as derived from CTFFIND4 (red) or CTFFIND5 (black). (**b**) Data for another tilt series plotted as described for (**a**). The estimated stage tilt axis is 179.8°, the estimated specimen pre-tilt is –21.9° with a tilt axis of 183.8°. This is consistent with this grid being inserted in the opposite orientation as the grid shown in (**a**), but still with a rough alignment of milling direction and tilt axis.

steps, respectively, using the locally fitted Thon ring patterns to score each pair of tilt axis and angle, followed by local refinement of tilt axis, angle, and average defocus.

Finally, an average tilt-corrected power spectrum at a tile size requested by the user (usually 512×512 pixels) is calculated for diagnostic purposes and for further refinement of CTF parameters. The tilt correction is designed to remove most of the Thon ring blurring due to the defocus variation across the image. To minimize ring blurring, the power spectrum from each tile is adjusted according to its local average defocus, $\Delta f_{average}$, by magnifying it by a factor $m$ with

$$m = \sqrt{\Delta f_{local}/\Delta f_{average}} \qquad (1)$$

Since $\Delta f_{local}$ will assume values across the image that are both smaller and larger than $\Delta f_{average}$, $m$ will assume values smaller and larger than 1. The magnification/demagnification of the power spectrum compensates for the contraction/expansion of the Thon rings due to the local defocus change and produces approximately constant Thon ring patterns that can be averaged without losing the pattern (*Figure 1b*). The compensation will have a small error if the spherical aberration is not zero. However, this error is sufficiently small to not visibly affect the Thon rings in the average (*Figure 2*).

## Verification of tilt estimation using tilted aquaporin crystals

To test the robustness and accuracy of the new fitting algorithm, the defocus and sample tilts of aquaporin 2D crystals (*Murata et al., 2000*) were estimated using a search range from 5000 Å to 50000 Å and a 100 Å step, low- and high-resolution limits of 30 Å to 5 Å, respectively, and a box size for the final power spectrum of 512 pixels. The estimated tilt angle $\theta$ and axis direction $\phi$ were compared with the values obtained by 2D crystallographic processing (*Mindell and Grigorieff, 2003*).

## Verification of tilt estimation using tilt series

Lamellae prepared from ER-HoxB8 cells were imaged using a Titan Krios 300 keV TEM controlled by SerialEM (*Mastronarde, 2005*). For each dataset, an initial exposure was taken with a magnification of 64,000, resulting in a pixel size of 1.6 Å and an exposure of 30 e⁻/Å². This was followed by the acquisition of a tilt series at a magnification of 48,000, resulting in a pixel size of 2.087 Å. A total of 35 tilt images at a tilt interval of 3° were collected from –51° to 51°, relative to the milling angle, using a grouped dose-symmetric scheme (*Hagen et al., 2017*). The exposure per tilt was 3 e⁻/Å², resulting in a total exposure of 105 e⁻/Å².

Lamellae prepared by FIB milling usually exhibit a pre-tilt with respect to the grid surface due to the stage tilt in the FIB instrument. In the microscope, the direction of this pre-tilt will generally not line up with the goniometer tilt axis. For the alignment of a tomogram recorded from such a lamella, the relative orientation of these two axes will have to be determined, together with the precise amount of pre-tilt. We wrote a new *cis*TEM (*Grant et al., 2018*) program, called fit_tilt_model, to read the tilt angles and axes determined for each image in a tomographic tilt series and fit them to a model incorporating a pre-tilt and a single tomographic tilt axis. Using a rotation matrix $R_0$ to represent the pre-tilt and rotation matrices $R_{tom}^i$ to represent the tomographic tilt angles and axis read from the microscope, the overall sample orientations are given by

$$R^i = R_{tom}^i \times R_0 \qquad (2)$$

$R_0$ and $R_{tom}^i$ are calculated from the tilt angles $\theta$ and axes $\phi$ as

$$R = \begin{bmatrix} \cos{(\phi)}^2 + \sin{(\phi)}^2 \cos{(\theta)} & \cos{(\phi)} \sin{(\phi)} \left(1 - \cos{(\theta)}\right) & \sin{(\phi)} \sin{(\theta)} \\ \cos{(\phi)} \sin{(\phi)} \left(1 - \cos{(\theta)}\right) & \cos{(\phi)}^2 \cos{(\theta)} + \sin{(\phi)}^2 & -\cos{(\phi)} \sin{(\theta)} \\ -\sin{(\phi)} \sin{(\theta)} & \cos{(\phi)} \sin{(\theta)} & \cos{(\theta)} \end{bmatrix} \qquad (3)$$

Using the tilt information obtained with CTFFIND5, we now have a set of rotation matrices $R^i$, and together with the rotation matrices read from the microscope, $R_{tom}^i$, we can calculate a set of pre-tilt estimates $R_0^i$ from *Equation 2*. To determine the best overall pre-tilt $R_0$, we determine the plane-normal vectors $V_{norm}^i = \left[x, y, z\right]^T$ of the sample by applying $R_0^i$ to the vector $\left[0, 0, 1\right]^T$ (z-coordinate along the beam direction), followed by calculating their mean $V_{norm}^{mean} = \left[x_0, y_0, z_0\right]^T$ as the normal vector of the best overall pre-tilt estimate. By calculating the root mean squared deviation of the normal

vectors $V^i_{norm}$, outliers can be identified and excluded to further refine $V^{mean}_{norm}$. The pre-tilt can then be determined as:

$$\theta_0 = \begin{cases} \cos^{-1}(z_0) & x_0 \geq 0 \\ -\cos^{-1}(z_0) & x_0 < 0 \end{cases}$$
$$\phi_0 = \begin{cases} \tan^{-1}\left(-\dfrac{x_0}{y_0}\right) & y_0 \neq 0, \phi_0 \in [0°, 180°] \\ 90° & y_0 = 0 \end{cases} \tag{4}$$

To generate more reliable defocus and tilt estimates, the defocus search range and resolution fitting range can be adjusted according to the experimental tilt range and image quality. For our cryoEM samples, the low- and high-resolution limits were set to 50 Å to 10 Å, respectively, and the defocus search interval was set to be between ±10,000 and ±20,000 Å from the nominal defocus set during data collection.

## Sample thickness estimation

In CTFFIND5 we implemented a new $CTF_t$ model function, based on the $CTF$ function implemented in CTFFIND4 (*Rohou and Grigorieff, 2015*) and extended by the formula described by *McMullan et al., 2015*:

$$CTF_t\left(\lambda, g, \Delta f, C_s, \Delta\varphi, \omega_2, t\right) = \frac{1}{2}\left(1 - \text{sinc}\left(\xi\left(\lambda, g, t\right)\right)\cos\left(2\chi\left(\lambda, |g|, \Delta f, C_s, \Delta\varphi, \omega_2\right)\right)\right) \tag{5}$$

where $\chi$ denotes the phase shift as a function of the electron wavelength $\lambda$, the spatial frequency vector $|g|$, the objective defocus $\Delta f$, the spherical aberration $C_s$, the additional phase shift $\Delta\varphi$, and the fraction of amplitude contrast $\omega_2$. The modulation of the CTF due to sample thickness $t$ is described by the function $\text{sinc}(\xi)$ with $\xi$ defined as:

$$\xi\left(\lambda, g, t\right) = \pi\lambda g^2 t \tag{6}$$

This sinc modulation envelope (*Figure 3a*) attenuates the amplitude of the Thon rings with increasing spatial frequencies and produces nodes where the apparent amplitude is zero (*Tichelaar et al., 2020*). After the first node the Thon rings appear 'inverted' compared to the CTFFIND4 CTF model, with maxima in the Thon rings appearing where the CTF model is 0. This causes the quality of fit indicator to rapidly decrease at the first node in CTFFIND4. The location of the first node is marked with a # in *Figure 3a*.

If a user requests sample thickness estimation, the program will first fit the $CTF$ model function as implemented in CTFFIND4. If the user also requested tilt estimation the tilt-corrected power spectrum will be used for all subsequent steps. Initially, the 'goodness-of-fit' resolution $g_{GoF}$ will be used as an estimate for the frequency of the first node of the $CTF_t$ function, which occurs when the sinc term in *Equation 5* becomes 0 for the first time at $\xi = \pi$. By setting $\xi = \pi$ in *Equation 6*, we can obtain an estimate of the sample thickness $t$, from $g_{GoF}$:

$$t = \frac{1}{\lambda g^2_{GoF}} \tag{7}$$

If the option 'Brute-force 1D fit' is selected, CTFFIND5 will further refine $t$ and $\Delta f$ by calculating the normalized cross-correlation between the radial average of the power spectrum, corrected for astigmatism by equiphase averaging (EPA) as described in *Zhang, 2016*, and $CTF_t$, searching systematically for the best combination of $t$ in the range of 50–400 nm in 10 nm steps, and $\Delta f$ in the range of ±200 nm from the previously fitted value, also in 10 nm steps.

Finally, if the option '2D-refinement' is selected, CTFFIND5 will optimize $t$, $\Delta f_1$, $\Delta f_2$, and $\omega$ using the same conjugate gradient algorithm used in CTFFIND4 and the normalized cross-correlation between $CTF_t$ and the 2D power spectrum as a scoring function.

After the optimal values for $t$ and $\Delta f$ have been obtained the 'goodness-of-fit' cross-correlation is recalculated using $CTF_t$, with a frequency window that is 1.5 times larger than in CTFFIND4 to avoid the drop-off in the node regions of $CTF_t$. For visualization, the astigmatism-corrected 1D power

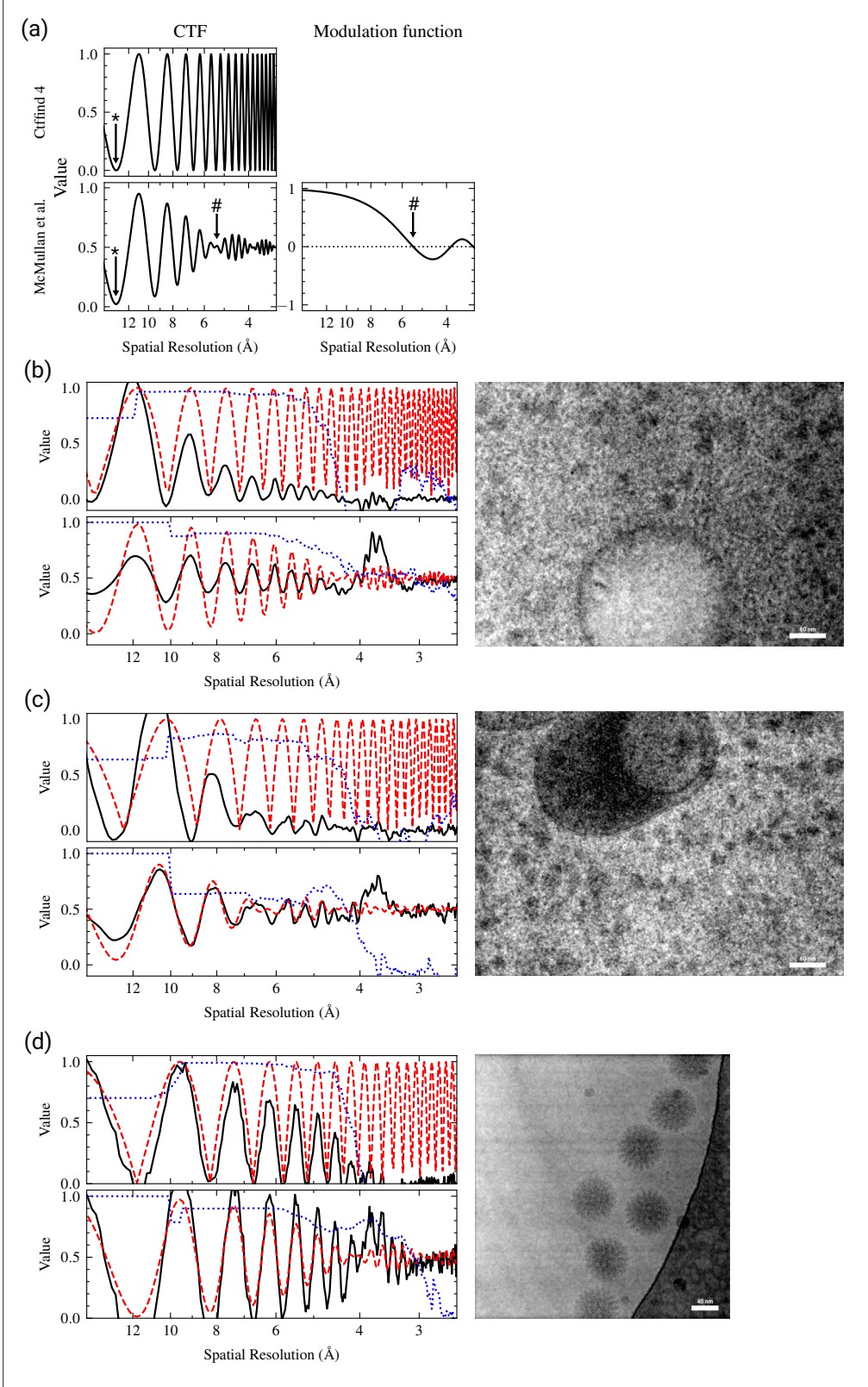

**Figure 3.** Sample thickness estimation by fitting Thon ring patterns. (**a**) Comparison of the contrast transfer function (CTF) model used in CTFFIND4, and after applying the modulation function (right) described by *McMullan et al., 2015*. A star symbol (*) denotes the position of the first zero in the CTF and a pound symbol (#) denotes the position of the first node. (**b–d**) Representative examples of Thon ring fitting in micrographs

*Figure 3 continued on next page*

*Figure 3 continued*

without (top-left graph) and with (bottom-left graph) thickness estimation. The micrograph is shown to the right. Each graph shows that equiphase averaging (EPA) of the power spectrum in solid black lines, the fitted CTF model in dashed red lines, and the goodness of fit indicator as a dotted blue lines. (**b**) Thon ring fitting of a micrograph taken from a focused ion beam (FIB) milling-derived lamella. The tilt of the specimen was estimated to be 12.3°. When fitting without thickness estimation the estimated parameters were $\Delta f_1 = 10492\text{Å}, \Delta f_2 = 10342\text{Å}, \alpha = 81.2°$. When taking sample thickness into account the estimated parameters were $\Delta f_1 = 10481\text{Å}, \Delta f_2 = 10286\text{Å}, \alpha = 69.6°, t = 969\text{Å}$. The estimated fit resolution was 4.6Å and 3.4Å without and with sample thickness estimation, respectively. (**c**) Thon ring fitting of a micrograph taken from a FIB milling-derived lamella. The tilt of the specimen was estimated to be 6.7°. When fitting without thickness estimation the estimated parameters were $\Delta f_1 = 8002\text{Å}, \Delta f_2 = 7717\text{Å}, \alpha = 73.4°$. When taking sample thickness into account the estimated parameters were $\Delta f_1 = 8549\text{Å}, \Delta f_2 = 8343\text{Å}, \alpha = 63.3°, t = 2017\text{Å}$. The estimated fit resolution was 4.3Å and 4.2Å without and with sample thickness estimation, respectively. (**d**) Thon ring fitting of a micrograph taken from plunge frozen rotavirus double-layered particles (*Grant and Grigorieff, 2015*). When fitting without thickness estimation the estimated parameters were $\Delta f_1 = 7027\text{Å}, \Delta f_2 = 6808\text{Å}, \alpha = -20.3°$. When taking sample thickness into account the estimated parameters were $\Delta f_1 = 7027\text{Å}, \Delta f_2 = 6808\text{Å}, \alpha = -22.9°, t = 850\text{Å}$. The estimated fit resolution was 4.2Å and 3.2Å without and with sample thickness estimation, respectively.

spectrum is displayed as oscillating around 0.5, as opposed to the default procedure in CTFFIND4, where the minima are scaled to 0. This is done to allow a better visual comparison to the $CTF_t$ model.

## Verification of sample thickness estimation using the Beer-Lambert law

We used 655 micrographs collected from one lamella of ER-HoxB8 cells (dataset Lamella $_{EUC}1$ from *Elferich et al., 2022*). For each micrograph we calculated $ln\left(\frac{I}{I_0}\right)$, where $I$ was the sum of all pixels in the illuminated area of the movie and $I_0$ was the average of this sum for 45 micrographs collected over vacuum with the same energy filter settings. This value is expected to have a linear relationship with the thickness of the sample consistent with the Beer-Lambert law (*Rice et al., 2018*; *Yan et al., 2015*):

$$ln\left(\frac{I}{I_0}\right) = \frac{1}{\kappa}t \tag{8}$$

where $\kappa$ is the apparent mean free path for inelastic scattering.

We then used CTFFIND5 to estimate the thickness $t$ of each micrograph using the 'Brute-force 1D fit' and '2D-refinement' setting, low- and high-resolution limits set to 30 Å and 5 Å, defocus search range set between 500 nm and 5000 nm, and low- and high-resolution limits for thickness

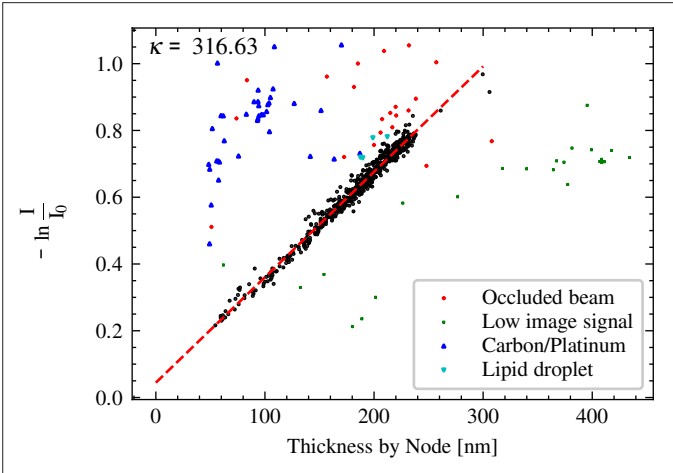

**Figure 4.** Validation of sample thickness estimation in CTFFIND5 by comparing the estimates to the intensity attenuation by the zero-loss energy filter. An estimation of the linear relationship using the RANSAC algorithm results in a slope of 1/316.6 nm and an *x*-axis intercept at –14 nm (red dashed line). Data points that were labeled as outliers by the RANSAC algorithm were manually inspected and color-coded according to visual inspection of the micrographs.

estimation set to 10 Å and 3 Å. We used a 'RANSAC' algorithm as implemented by the scikit-learn Python package (*Pedregosa et al., 2011*) to fit a linear model to the relationship of $ln\left(\frac{I}{I_0}\right)$ and $t$, while rejecting outliers. We then manually inspected every outlier of the model fit and categorized the reason for the discrepancy into 'occluded beam' (either from contamination or the edges of the lamella), 'low image signal' (in most cases exposures containing no cellular features), 'carbon/platinum', and 'lipid droplet' (see *Figure 4*).

## Verification of sample thickness estimation using tomography

The same tilts series described in the section 'Verification of tilt estimation using tilt series' were used here. For tomographic reconstruction, tilt movie frame motion correction was performed using SerialEM (*Mastronarde, 2005*), and tilt series were aligned using the IMOD software package (version 4.11, *Mastronarde and Held, 2017*). For coarse alignment, a high-frequency cutoff radius of 0.15 was used. A fiducial model was generated using patch tracking with patches of 450×450 pixels and a fractional overlap of patches of 0.33×0.33. High-tilt frames were omitted while generating the fiducial model. Robust fitting with a tuning factor of 1 was used for fine alignment. After computing the alignment, the fiducial model was edited by removing unreliable patches, and then alignments were re-computed. The edited models with the lowest residual mean errors and standard deviations were used for fine alignment. Tomogram positioning was used to correct the tilt angle offset. Fully aligned stacks were generated with a binning factor of 4, resulting in a tomogram pixel size of 8.3 Å. Tomograms were reconstructed using the SIRT-like filtering option in IMOD (*Mastronarde, 1997*; *Mastronarde and Held, 2017*) and manually inspected. The tomograms were back-projected along the *y*-axis using a homemade script, generating a small set of *XZ* projections. Thickness measurements on the projected central slices were performed using the display program included with the *cis*TEM software package (*Grant et al., 2018*).

## CTF correction of medium-magnification lamella images

The CTF of the representative medium-magnification image with a pixel size of 40 Å was estimated using CTFFIND5 with the following parameters: defocus range: 1,000,000–4,000,000 Å; search step 50,000 Å; low- and high-resolution limits: 400 Å and 80 Å. We then used the program apply_ctf, included with *cis*TEM, to flip the phases according to the estimated CTF. We furthermore implemented the Wiener-like filter described in *Tegunov and Cramer, 2019*, in apply_ctf to produce the image shown in Figure 6d.

## Benchmarking CTFFIND5 runtimes

CTFFIND5 runtimes were measured using three representative micrographs (Table 2). As a baseline measurement, CTFFIND5 was run without estimation of tilt and sample thickness enabled. Then runtime was measured enabling either one of these options or both. Every test was repeated four times and the average and standard deviation of the last three runs are reported, to minimize the contribution of hard-drive speed. The tests were performed on a single core of an Intel Core i9-12900KF CPU.

## Results

### Tilt estimation

We tested the tilt correction for the Thon rings on a representative micrograph taken from a cryo-FIB-milled lamella. As expected, the correction results in the observation of Thon rings at higher spatial resolution (*Figure 1c*). In this example, correcting for the estimated moderate tilt of 12.3° improved the highest resolution at which a reasonable fit could be obtained from 5.9 Å to 4.6 Å. The power spectrum also appears less noisy, which can be attributed to some low-pass filtering that occurs with the interpolation of the Thon ring patterns of individual tiles to perform the tilt correction.

   To test the performance of the new CTFFIND5 sample tilt estimation, we used a dataset of images of tilted aquaporin crystals that were also used to benchmark the original CTFTILT implementation (*Mindell and Grigorieff, 2003*; *Murata et al., 2000*). *Table 1* compares the tilt information of the samples obtained from crystallographic analysis and the estimates obtained using CTFFIND5. Overall, the results of CTFFIND5 agree well with the aquaporin crystals information. The average discrepancy was 1.9° for the tilt axis direction and 1.5° for the tilt angle.

**Table 1.** Comparison of CTFFIND5 estimation of sample tilt with crystallographic analysis.

| Image | Axis angle $\varphi$ | | | Tilt angle $\theta$ | | |
|---|---|---|---|---|---|---|
| | Crystallog. | CTFFIND5 | $\Delta\varphi$ | Crystallog. | CTFFIND5 | $\Delta\theta$ |
| 530394 | 93.28 | 94.98 | −1.7 | 19.6 | 20.69 | −1.09 |
| 530419 | 109.78 | 106.51 | 3.27 | 18.66 | 16.04 | 2.62 |
| 530430 | 104.38 | 101.13 | 3.25 | 21.32 | 20.37 | 0.95 |
| 530444 | 98.39 | 97.62 | 0.77 | 20.72 | 20.88 | −0.16 |
| 660027 | 99.68 | 102.34 | −2.66 | 19.4 | 22.39 | −2.99 |
| 540149 | 94.45 | 85.84 | 8.61 | 43.08 | 44.59 | −1.51 |
| 540291 | 96.16 | 98.1 | −1.94 | 45.11 | 40.68 | 4.43 |
| 540302 | 93.98 | 93.39 | 0.59 | 44.7 | 44.21 | 0.49 |
| 540313 | 95.34 | 95.13 | 0.21 | 44.03 | 46.49 | −2.46 |
| 660183 | 97.69 | 97.27 | 0.42 | 48.13 | 48.99 | −0.86 |
| 550069 | 90.08 | 92.55 | −2.47 | 60.46 | 60.83 | −0.37 |
| 550089 | 91.48 | 92.04 | −0.56 | 60.5 | 60.72 | −0.22 |
| 660291 | 93.23 | 92.19 | 1.04 | 57.59 | 59.19 | −1.60 |
| 660421 | 89.32 | 89.06 | 0.26 | 61.36 | 60.01 | 1.35 |
| 680341 | 89.67 | 90.02 | −0.35 | 58.68 | 59.62 | −0.94 |
| 530345 | N/A | 108.6 | | 0 | 0.84 | −0.84 |
| 530356 | N/A | 231.17 | | 0 | 1.93 | −1.93 |
| 530358 | N/A | 56.58 | | 0 | 1.29 | −1.29 |
| 530375 | N/A | 3.21 | | 0 | 0.79 | −0.79 |
| 530378 | N/A | 67.6 | | 0 | 2.17 | −2.17 |

To test whether CTFFIND5 would be able to correctly assign tilt axis and angle for tilt series data, we analyzed two tilt series from different grids of lamellae prepared by cryo-FIB milling from mouse neutrophil-like cells (*Elferich et al., 2022*). We then plotted the estimated values for tilt axis and angle as a function of nominal stage tilt (*Figure 2*). The estimated tilt angle shows a roughly linear relationship with the nominal stage tilt, but since CTFFIND5 reports only positive tilt angles the overall plot has a V-shape. The estimated tilt axis angle is approximately constant at high tilts but changes by about 180° at 0° estimated tilt, again due to the convention enforced by CTFFIND5. Notably, in both examples there is an offset of about 20° between nominal and estimated tilts, which is due to the pre-tilt of the specimen caused by FIB milling at a shallow angle. To quantify and delineate both the tilt axis direction of the microscope and the pre-tilt of the specimen we fit all values to a model as described in Methods (*Figure 2*). The fitting resulted in an estimated tilt axis angle of 178.2° and 179.8°, respectively, which is consistent with the SerialEM calibration of 178.4° and 176.3° for the stage tilt axis. The estimated pre-tilt values were 20.6 °and –21.9°, consistent with a FIB-milling angle of 20° and opposite orientation of the grids relative to the milling direction. The pre-tilt axis angles were estimated as 171.8° and 183.8°, which is consistent with the error expected from manually aligning the milling direction when inserting grids into the microscope.

To estimate the accuracy of the tilt estimation in tilt series, we calculated the mean absolute difference between the tilt and axis-angle estimates and the fitted model, excluding the axis-angle estimates at tilt angles under 5°. For the first tilt series we obtained accuracy estimates of 2.08° and 2.58° for tilt and axis angles, respectively. In the second tilt series the accuracy estimates were 3.95° and 9.47°. In both cases the accuracy was lower than for the tilted aquaporin crystals, presumably due to the relatively short exposure of each micrograph in the tilt series. However, the substantially higher

mean differences in the second tilt series suggest that the accuracy is highly dependent on the quality of the underlying data.

## Sample thickness estimation

Even after correcting for sample tilt we found that for FIB-milled samples we often could observe Thon ring-like modulation in the power spectrum at higher resolution than suggested by the goodness-of-fit estimate (*Figure 3b*, top plot). These modulations are out of phase with the predicted modulations, as described by *McMullan et al., 2015*, and *Tichelaar et al., 2020*. We therefore implemented an extension of the CTF model as described by *McMullan et al., 2015*; *Figure 3a*. For some images we found that the thickness could be well estimated by assuming that the goodness-of-fit resolution estimate obtained using the old model implemented in CTFFIND4 corresponds to the first node in the modulation function, according to *Equation 7*. With our new model, estimated CTF parameters were very similar to those from CTFFIND4, but the fit in CTFFIND5 extended to higher resolution (*Figure 3b*).

In other images, mostly with defocus values under 1 μm and with a sample thickness over 200 nm, CTFFIND4 could fit the power spectrum before and after the first node using the old CTF model, with some deviations between the fit and the power spectrum (*Figure 3c*). Fitting the power spectrum with the new model in CTFFIND5 resulted in substantially different estimated CTF parameters and an improved fit, even though the goodness-of-fit estimation did not change. We also found that for some single-particle datasets of large particles, such as the rotavirus double-layered particle (*Grant and Grigorieff, 2015*), we could use CTFFIND5 to estimate the thickness of the sample, which resulted in CTF fits to higher resolution (*Figure 3d*).

Based on these results we conclude that CTFFIND5 will provide more accurate CTF parameters for images of thick samples, such as those generated from FIB milling. In addition, the fit provides a direct readout of the specimen thickness, which is important for judging specimen quality and the potential for high-resolution information that can be recovered from these images.

## Estimating the accuracy of sample thickness estimation using the Beer-Lambert law on energy filtered data

CryoEM is frequently performed using an energy filter to remove inelastically scattered electrons. The fraction of inelastically scattered electrons can be described by the Beer-Lambert law, which states that the fraction of electrons removed from the image is proportional to the thickness of the sample. The apparent mean free path for electron scattering has been experimentally determined for common cryoEM conditions (*Rice et al., 2018*). To test whether thickness estimation in CTFFIND5 is consistent with this method, we used a dataset of 655 exposures of a lamella of ER-HoxB8 cells collected using the DeCo-LACE approach (*Elferich et al., 2022*). We used CTFFIND5 to estimate the thickness $t$ of every exposure and plotted $-ln\left(\frac{I}{I_0}\right)$ against $t$ (*Figure 4*). Fitting the data to a linear model described in Methods (*Equation 8*), we found that 568 out of 655 exposures followed closely a linear relationship with a mean free path $\kappa$ of 317 nm. Manual inspection of images that did not follow this linear relationship revealed that they either contained visible ice contamination, platinum deposits, or they were collected over ice without cellular features and displayed weak Thon rings. The value of $\kappa$ is consistent with the value found by *Rice et al., 2018*, even though our dataset was collected without an objective aperture. The x-axis intercept of the linear model was –14.1 nm, meaning that the node position systematically predicts a smaller thickness than predicted by the Beer-Lambert law. This discrepancy is further discussed in the next section. To estimate the accuracy of the sample thickness determined by CTFFIND5, we calculated the mean absolute difference to the linear model, which was 4.8 nm. These data suggest that sample thickness determination using node fitting is an alternative to using the Beer-Lambert law that has the advantage of not relying on the constant $\kappa$ and the intensity $I_0$, both of which might not be readily available. Also, the two approaches are complementary as they rely on orthogonal mechanisms.

## Estimating the accuracy of sample thickness estimation using tomography

We used a dataset of seven micrographs collected from lamellae of ER-HoxB8 cells together with tilt series collected afterward from the same locations to verify the accuracy of the thickness estimates obtained using CTFFIND5. We used CTFFIND5 to estimate the thickness ($t_{CTFFIND}$) for every

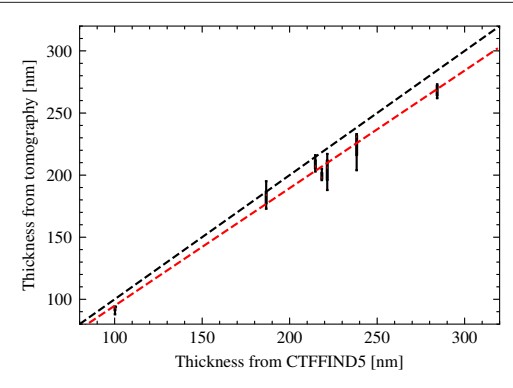

**Figure 5.** Validation of sample thickness estimation in CTFFIND5 by tomography. The distribution of thickness measurements in seven tomograms are shown as box plots with the median indicated by a red line. For each tomogram, the thickness was measured in three different places. The position on the *x*-axis corresponds to the thickness estimate by CTFFIND5. The black dashed line indicates identity. The red dashed line indicates the result of a linear fit.

location and compared it with the thickness estimated and compared it with the thickness estimated from the tomogram reconstructed from the tilt series ($t_{TOMO}$). We measured $t_{TOMO}$ by manually estimating the distance between the surfaces of the lamella in three different positions. When we plotted $t_{CTFFIND}$ against $t_{TOMO}$ we found that the values were highly correlated, but $t_{TOMO}$ was consistently smaller than $t_{CTFFIND}$ (*Figure 5*). A linear fit revealed a slope of 0.95 and a *y*-axis intercept of 0.12 nm. This means that the CTFFIND5 thickness estimate is on average 1.05× higher than the thickness estimated by tomography. *Tichelaar et al., 2020*, also report that estimating the thickness from the CTF nodes resulted in values roughly 1.1× higher than estimated by tomography. The reasons for the systematic discrepancies between thicknesses estimated by CTFFIND5 and estimates based on the Beer-Lambert law and tomography are unclear, but since they are small and CTFFIND5 estimates lie in between the other two estimates, they will provide comparable information.

## CTF estimation and correction assists biological interpretation of intermediate-magnification lamella images

During data collection of cryoEM data in cells, the operator frequently relies on images taken at low magnification to select areas of interest and establish their biological context. The pixel size of these images is usually about 40 Å, with a defocus of about 200 µm. This produces strong contrast from biological membranes but can sometimes also lead to substantial fringes near these membranes (*Figure 6a*). We found that a simple CTF correction based on CTFFIND defocus estimates obtained from the overview images can reduce these fringes (*Figure 6b*). A simple CTF correction can be done using the program apply_ctf, included with *cis*TEM, by phase flipping according to the fitted CTF (*Figure 6c*). However, we found that including a Wiener filter-based amplitude correction described by *Tegunov and Cramer, 2019*, produces a more naturally looking image that might be best suited to recognize cellular features (*Figure 6d*).

## CTFFIND5 runtimes

To gauge the ability of CTFFIND5 to provide real-time feedback during cryoEM data collection, we measured its runtime on three representative micrographs (*Table 2*). Without estimation of tilt or sample thickness CTFFIND5 performed CTF estimation roughly within a second. Estimation of the sample thickness adds roughly half a second to the runtime, therefore allowing CTF estimation within a timeframe comparable to typical exposure times. Estimation of the tilt on the other hand increased runtimes substantially to the order of several minutes, due to the exhaustive search of potential tilts over hundreds of power spectra. While these runtimes are substantially slower than cryoEM data acquisition, near real-time estimation can be achieved by using multiple CPU cores. Furthermore, optimization of the number of tiles used, better search algorithms, or implementations employing GPUs could increase the speed to the point where real-time estimation is more feasible.

## Conclusion

The new features implemented in CTFFIND5 improve CTF estimation from the power spectra of cryoEM micrographs where assumptions made in its predecessor, CTFFIND4, namely a thin and untilted sample, do not hold. The tilt of the sample is estimated by fitting the CTF to the power spectra calculated from small patches across the image, similar to other software including CTFTilt (*Mindell and Grigorieff, 2003*), Ctfplotter (*Mastronarde, 2024*; *Xiong et al., 2009*), goCTF (*Su,*

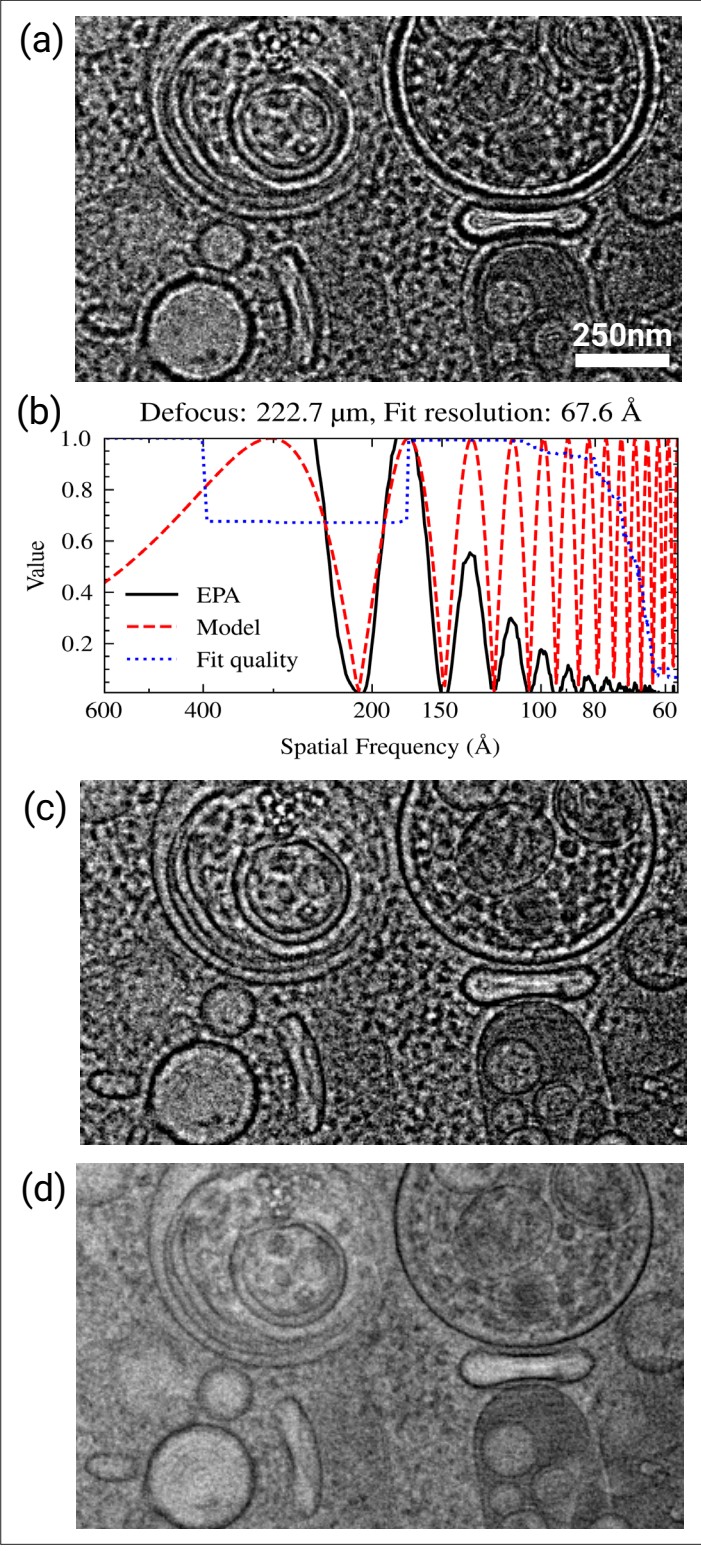

**Figure 6.** Contrast transfer function (CTF) correction of medium-magnification overviews. (**a**) Representative area of a micrograph of a cellular sample at a pixel size of 40 Å without CTF correction. (**b**) Fit of the power spectrum of the micrograph shown in panel (a) CTF model. (**c–d**) The same micrograph as shown in panel (**a**) after CTF correction by phase flipping (**c**) or with a Wiener-like filter (**d**). The custom fall-off parameter was set to 1.3 and the custom strength parameter was set to 0.7.

**Table 2.** Runtime of CTFFIND5 on representative micrographs.

| Micrograph | | 1 | 2 | 3 |
|---|---|---|---|---|
| Image properties | | | | |
| Image size | | 4070×2892 | 2880×2046 | 4746×3370 |
| Pixel size (Å) | | 1.5 | 4.175 | 2.5 |
| Runtime (s) | | | | |
| Tilt | Thickness | | | |
| – | – | 0.9±0.1 | 0.7±0.1 | 1.7±0.1 |
| + | – | 39.0±0.2 | 208±1 | 173.4±0.1 |
| – | + | 1.4±0.1 | 1.3±0.1 | 2.4±0.1 |
| + | + | 39.5±0.1 | 209±1 | 173.0±0.1 |

*2019*), Bsoft (*Heymann and Belnap, 2007*), and Warp (*Tegunov and Cramer, 2019*). After estimation of the sample tilt, a tilt-corrected power spectrum is produced that exhibits stronger Thon rings at higher resolution.

To take into account the modulation of the power spectra by thick samples (*McMullan et al., 2015*; *Tichelaar et al., 2020*), we fit a modified CTF model, which increases the resolution of the fitted regions of the spectra and provides a readout of the sample thickness. While the low exposures (3–5 $e^-/Å^2$) typically used in electron cryo-tomography often preclude fitting of sample thickness from power spectra of individual images in the tilt series, we demonstrate that this works reliably for higher exposures (~30 $e^-/Å^2$) typically used for 2D template matching (2DTM) (*Lucas et al., 2021*; *Rickgauer et al., 2017*) and in situ single-particle analysis (*Cheng et al., 2023*).

We mainly developed these improvements for 2DTM applied to in situ samples, e.g., prepared by cryo-FIB milling. Since the sample thickness and sample tilt are apparent from a single exposure, as opposed to a reconstructed tomogram, CTFFIND5 made it possible to quantify these parameters for every exposure. This is important information to judge the quality of the lamella before performing time-intensive further processing (*Lucas and Grigorieff, 2023*; *Tuijtel et al., 2024*). Furthermore, since the quality-of-fit estimation of CTFFIND5 goes to higher resolution in tilted and thick samples (*Figures 1c and 3b–d*), it is a better indicator of micrograph quality than the same metric in CTFFIND4.

Even though we used tilt series data to benchmark tilt estimation in CTFFIND5, there are certain drawbacks in its use for tilt series data. In contrast to software optimized for tilt series data, such as Ctfplotter (*Mastronarde, 2024*), CTFFIND5 uses no prior information for tilt axis angle and sample tilt, resulting in reduced performance and also potential lower accuracy. Furthermore, there is no option to pool data from several tilt angles to provide a more accurate defocus or astigmatism estimation. However, this also means that CTFFIND5 together with fit_tilt_model might be useful in troubleshooting angle conventions in tilt series data. Also, existing pipelines using CTFFIND4 might benefit from the tilt correction in the power spectrum calculation introduced in CTFFIND5, which results in more robust defocus estimation at high tilt (*Figure 2*).

While most single-particle samples are thin and flat and therefore do not benefit from CTFFIND5, there might be some samples, such as viral particles (*Figure 3d*), where direct estimation of the ice thickness is valuable. However, even in these cases the defocus reported by CTFFIND5 is an average for the whole micrograph and per-particle CTF refinement, as implemented in *cis*TEM (*Grant et al., 2018*), RELION (*Zivanov et al., 2018*), or CryoSPARC (*Punjani et al., 2017*; *Zivanov et al., 2020*), will still be required to actually recover high-resolution information during averaging. The ability to measure sample tilt might be useful for approaches where a tilted stage is used to overcome preferred particle orientation (*Tan et al., 2017*), as initial per particle defocus values can be derived from the tilt as reported by CTFFIND5 and the particle positions.

In summary, the improvements implemented in CTFFIND5 result in more accurate CTF estimation of thick and tilted samples and provide valuable information about the samples to the microscopist.

## Acknowledgements

We would like to thank members of the Grigorieff lab for testing CTFFIND5 and helpful discussion. We furthermore thank Benjamin Himes for helpful discussions and code review within *cis*TEM. We thank Alexis Rohou for comments on the manuscript. LK and NG gratefully acknowledge funding from the Chan Zuckerberg Initiative, grant #2021-234617 (5022).

## Additional information

### Competing interests

Nikolaus Grigorieff: Reviewing editor, *eLife*. The other authors declare that no competing interests exist.

### Funding

| Funder | Grant reference number | Author |
| --- | --- | --- |
| Howard Hughes Medical Institute | | Nikolaus Grigorieff |
| Chan Zuckerberg Initiative | #2021-234617 (5022) | Lingli Kong Nikolaus Grigorieff |

The funders had no role in study design, data collection and interpretation, or the decision to submit the work for publication.

### Author contributions

Johannes Elferich, Conceptualization, Resources, Data curation, Software, Formal analysis, Supervision, Validation, Investigation, Visualization, Methodology, Writing - original draft, Project administration, Writing - review and editing; Lingli Kong, Data curation, Software, Formal analysis, Validation, Investigation, Visualization, Methodology, Writing - original draft, Writing - review and editing; Ximena Zottig, Data curation, Formal analysis, Validation, Investigation, Visualization, Methodology, Writing - original draft, Writing - review and editing; Nikolaus Grigorieff, Conceptualization, Software, Formal analysis, Supervision, Funding acquisition, Investigation, Methodology, Writing - original draft, Project administration, Writing - review and editing

### Author ORCIDs

Johannes Elferich  https://orcid.org/0000-0002-9911-706X
Lingli Kong  https://orcid.org/0000-0002-5808-2649
Ximena Zottig  https://orcid.org/0000-0002-2344-5163
Nikolaus Grigorieff  https://orcid.org/0000-0002-1506-909X

Reviewer #1 (Public review): https://doi.org/10.7554/eLife.97227.3.sa1
Reviewer #2 (Public review): https://doi.org/10.7554/eLife.97227.3.sa2
Author response https://doi.org/10.7554/eLife.97227.3.sa3

## Additional files

### Supplementary files

• MDAR checklist

### Data availability

The images of tilted aquaporin crystals were previously published (*Murata et al., 2000*) and are available at https://grigoriefflab.umassmed.edu/tilted_aquaporin_crystals. The untilted exposures

of ER-HOXB8 cells are available at EMPIAR (EMPIAR-11063). The tomograms and tilt series from ER-HOXB8 cells have been deposited to EMDB (EMD-43419, EMD-43420, EMD-43424, EMD-43425, EMD-43427, EMD-43428, EMD-43429) and EMPIAR (EMPIAR-11854), respectively. The source code for CTFFIND5 is available at https://github.com/timothygrant80/cisTEM/tree/ctffind5 (copy archived at *Rohou et al., 2024*) and binaries for most Linux distributions can be downloaded at https://cistem.org/development.

The following datasets were generated:

| Author(s) | Year | Dataset title | Dataset URL | Database and Identifier |
|---|---|---|---|---|
| Elferich JE, Kong LK, Zottig XZ, Grigorieff NG | 2024 | CTFFIND5 provides improved insight into quality, tilt and thickness of TEM samples | https://www.ebi.ac.uk/empiar/EMPIAR-11854/ | ArrayExpress, EMPIAR-11854 |
| Elferich J, Kong L, Zottig X, Grigorieff N | 2024 | Tomogram 1 - thickness measurement | https://www.ebi.ac.uk/emdb/EMD-43419 | Electron Microscopy Data Bank, EMD-43419 |
| Elferich J, Kong L, Zottig X, Grigorieff N | 2024 | Tomogram 2 - Thickness measurement | https://www.ebi.ac.uk/emdb/EMD-43420 | Electron Microscopy Data Bank, EMD-43420 |
| Elferich J, Kong L, Zottig X, Grigorieff N | 2024 | Tomogram 3 - Thickness measurement | https://www.ebi.ac.uk/emdb/EMD-43424 | Electron Microscopy Data Bank, EMD-43424 |
| Elferich J, Kong L, Zottig X, Grigorieff N | 2024 | Tomogram 4 - Thickness measurement | https://www.ebi.ac.uk/emdb/EMD-43425 | Electron Microscopy Data Bank, EMD-43425 |
| Elferich J, Kong L, Zottig X, Grigorieff N | 2024 | Tomogram 5 - Thickness measurement | https://www.ebi.ac.uk/emdb/EMD-43427 | Electron Microscopy Data Bank, EMD-43427 |
| Elferich J, Kong L, Zottig X, Grigorieff N | 2024 | Tomogram 6 - Thickness measurement | https://www.ebi.ac.uk/emdb/EMD-43428 | Electron Microscopy Data Bank, EMD-43428 |
| Elferich J, Kong L, Zottig X, Grigorieff N | 2024 | Tomogram 7 - Thickness measurement | https://www.ebi.ac.uk/emdb/EMD-43429 | Electron Microscopy Data Bank, EMD-43429 |

The following previously published dataset was used:

| Author(s) | Year | Dataset title | Dataset URL | Database and Identifier |
|---|---|---|---|---|
| Elferich JE, Schiroli GS, Scadden DS, Grigorieff NG | 2022 | Cryo-EM data and 2DTM results of entire sections of differentiated ER-HoxB8 cells | https://www.ebi.ac.uk/empiar/EMPIAR-11063/ | ArrayExpress, EMPIAR-11063 |

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
