## [Editor Report · eLife Assessment]

This **valuable** work presents the latest version of CTFFIND, which is the most popular software for determination of the contrast transfer function (CTF) in cryo-electron microscopy. CTFFIND5 estimates and considers acquisition geometry and sample thickness, which leads to improved CTF determination. The paper describes **compelling** evidence that CTFFIND5 finds better CTF parameters than previous methods, in particular for tilted samples (e.g. for cryo-electron tomography) or where thickness is an issue (e.g. cellular samples, or electron microscopy at low voltages).

---

## [Referee Report · Reviewer #1 (Public review)]

This work presents CTFFIND5, a new version of the software for determination of the Contrast Transfer Function (CTF) that models the distortions introduced by the microscope in cryoEM images. CTFFIND5 can take acquisition geometry and sample thickness into consideration to improve CTF estimation.

To estimate tilt (tilt angle and tilt axis), the input image is split into tiles and correlation coefficients are computed between their power spectra and a local CTF model that includes the defocus variation according to a tilted plane. As a final step, by applying a rescaling factor to the power spectra of the tiles, an average tilt-corrected power spectrum is obtained used for diagnostic purposes and estimate the goodness of fit. This global procedure and the rescaling factor resemble those used in Bsoft, Warp, etc, with determination of the tilt parameters being a feature specific of CTFFIND5 (and formerly CTFTILT). The performance of the algorithm is evaluated with tilted 2D crystals and tilt-series, demonstrating accurate tilt estimation in general.

CTFFIND5 represents the first CTF determination tool that considers the thickness-related modulation envelope of the CTF firstly described by McMullan et al. (2015) and experimentally confirmed by Tichelaar et al. (2020). To this end, CTFFIND5 uses a new CTF model that takes the sample thickness into account. CTFFIND5 thus provides more accurate CTF estimation and, furthermore, gives an estimation of the sample thickness, which may be a valuable resource to judge the potential for high resolution. To evaluate the accuracy of thickness estimation in CTFFIND5, the authors use the Lambert-Beer law on energy-filtered data and also tomographic data, thus demonstrating that the estimates are reasonable for images with exposure around 30 e/A2. While consideration of sample thickness in CTF determination sounds ideally suited for cryoET, practical application under the standard acquisition protocols in cryoET (exposure of 3-5 e/A2 per image) is still limited. In this regard, the authors are precise in the conclusions and clearly identify the areas where thickness-aware CTF determination will be valuable at present: in situ single particle analysis and in vitro single particle cryoEM of large specimens (e.g. viral particles).

In conclusion, the manuscript introduces novel methods inside CTFFIND5 that improve CTF estimation, namely acquisition geometry and sample thickness. The evaluation demonstrates the performance of the new tool, with fairly accurate estimates of tilt axis, tilt angle and sample thickness and improved CTF estimation. The manuscript critically defines the current range of application of the new methods in cryoEM.

---

## [Referee Report · Reviewer #2 (Public review)]

This paper describes the latest version of the most popular program for CTF estimation for cryo-EM images: CTFFIND5. New features in CTFFIND5 are the estimation of tilt geometry, including for samples, like FIB-milled lamellae, that are pre-tilted along a different axis than the tilt axis of the tomographic experiment, plus the estimation of sample thickness from the expanded CTF model described by McMullan et al (2015). The results convincingly show the added value of the program for thicker and tilted images, such as are common in modern cryo-ET experiments. The program will therefore have a considerable impact on the field.

Comments on revised version:

My comments have been addressed adequately.

---

## [Author Response]

The following is the authors’ response to the original reviews.

We thank the reviewers for their detailed comments. Several comments revolved around potential improvements in the 3D reconstructions that are obtained in later steps of the image processing pipelines for single-particle cryoEM and cryo-electron tomography. We have not investigated how our improvements in CTFFIND5 affect these downstream results and can therefore not make specific and quantitative statements in this regard. However, CTFFIND5 provided additional information about the sample that users will find useful (thickness, tilt) for selecting the data they would like to include in later processing, and how to process them. Furthermore, when the sample tilt of a thin specimen is known, local defocus estimates (e.g., per-particle defocus estimates) will be more accurate compared to estimates that ignore tilt information. In the following, we provide point-by-point responses to the reviewers’ comments.

**Reviewer #1 (Public Review):**
This work presents CTFFIND5, a new version of the software for determination of the Contrast Transfer Function (CTF) that models the distortions introduced by the microscope in cryoEM images. CTFFIND5 can take acquisition geometry and sample thickness into consideration to improve CTF estimation.To estimate tilt (tilt angle and tilt axis), the input image is split into tiles and correlation coefficients are computed between their power spectra and a local CTF model that includes the defocus variation according to a tilted plane. As a final step, by applying a rescaling factor to the power spectra of the tiles, an average tilt-corrected power spectrum is obtained and used for diagnostic purposes and to estimate the goodness of fit. This global procedure and the rescaling factor resemble those used in Bsoft, Warp, etc, with determination of the tilt parameters being a feature specific of CTFFIND5 (and formerly CTFTILT). The performance of the algorithm is evaluated with tilted 2D crystals and tiltseries, demonstrating accurate tilt estimation in some cases and some limitations in others. Further analysis of CTF determination with tilt-series, particularly showing whether there is accurate or stable estimation at high tilts, might be helpful to show the robustness of CTFFIND5 in cryoET.CTFFIND5 represents the first CTF determination tool that considers the thickness-related modulation envelope of the CTF firstly described by McMullan et al. (2015) and experimentally confirmed by Tichelaar et al. (2020). To this end, CTFFIND5 uses a new CTF model that takes the sample thickness into account. CTFFIND5 thus provides more accurate CTF estimation and, furthermore, gives an estimation of the sample thickness, which may be a valuable resource to judge the potential for high resolution. To evaluate the accuracy of thickness estimation in CTFFIND5, the authors use the Lambert-Beer law on energy-filtered data and also tomographic data, thus demonstrating that the estimates are reasonable for images with exposure around 30 e/A2. While consideration of sample thickness in CTF determination sounds ideally suited for cryoET, practical application under the standard acquisition protocols in cryoET (exposure of 3-5 e/A2 per image) is still limited. In this regard, the authors are honest in the conclusions and clearly identify the areas where thickness-aware CTF determination will be valuable at present: e.g. in situ single particle analysis and in vitro single particle cryoEM of purified samples at low voltages.In conclusion, the manuscript introduces novel methods inside CTFFIND5 that improve CTF estimation, namely acquisition geometry and sample thickness. The evaluation demonstrates the performance of the new tool, with fairly accurate estimates of tilt axis, tilt angle and sample thickness and improved CTF estimation. The manuscript critically defines the current range of application of the new methods in cryoEM.
**Reviewer #2 (Public Review):**
Summary:This paper describes the latest version of the most popular program for CTF estimation for cryo-EM images: CTFFIND5. New features in CTFFIND5 are the estimation of tilt geometry, including for samples, like FIB-milled lamellae, that are pre-tilted along a different axis than the tilt axis of the tomographic experiment, plus the estimation of sample thickness from the expanded CTF model described by McMullan et al (2015). The results convincingly show the added value of the program for thicker and tilted images, such as are common in modern cryo-ET experiments. The program will therefore have a considerable impact on the field.I have only minor suggestions for improvement below:Abstract: "[CTF estimation] has been one of the key aspects of the resolution revolution"-> This is a bit over the top. Not much changed in the actual algorithms for CTF estimation during the resolution revolution.

We have removed this statement in the abstract.

L34: "These parameters" -> Cs is typically given, only defocus (and if relevant phase shift) are estimated.

We have modified the introduction to reflect this. Page 3, L30-35

L110-116: The text is ambiguous: are rotations defined clockwise or counter-clockwise? It would be good to explicitly state what subsequent rotations, in which directions and around which axes this transformation matrix (and the input/output angles in CTFFIND5) correspond to.

Thank you for pointing this out. We have revised the Methods section, Page 4 L57-61, to explicitly define the convention for the tilt axis and tilt angle. We have also modified Fig. 1b to illustrate our convention for the tilt axis.

L129-130: As a suggestion: it would be relatively easy, and possibly beneficial to the user, to implement a high-resolution limit that varies with the accumulated dose on the sample. One example of this exists in the tomography pipeline of RELION-5.

We appreciate the suggestion. However, since CTFFIND5 currently has no concept of a tilt-series and treats every micrograph independently, this would not be trivial to implement. As detailed below, CTFFIND5 in its current form is not targeted toward tomography processing, but its features might be useful for its use in pipelines for tomography processing, such as RELION-5. We made this more explicit in the conclusion section. Page 16 L390-399

Substituting Eq (7) into Eq (6) yields ksi=pi, which cannot be true. If t is the sample thickness, then how can this be a function of the frequency g of the first node of the CTF function? The former is a feature of the sample, the latter is a parameter of the optical system. This needs correction.

We have rewritten the text describing equations 7 and 6 to avoid this confusion (Page 7, L146-153). The reviewer is right that inserting Eq. 7 into Eq. 6 yields ksi=psi, as in fact Eq. 7 is derived from Eq. 6, by substituting ksi=psi, since this describes the condition for the first node. Also, in this context, nodes in the CTF function refer to the places where the term sinc(ksi) becomes zero and therefore the CTF is apparently "flat". The frequency at which this occurs is sample-thickness dependent. As explained below, the previous version of our manuscript did not point out the difference between the first zero and first node in the power spectrum. We have amended Fig. 3a to make this difference clearer.

**Reviewer #3 (Public Review):**
In this manuscript, the authors detail improvements in the core CTFFIND (CTFFIND5 as implemented in cisTEM) algorithm that better estimates CTF parameters from titled micrographs and those that exhibit signal attenuation due to ice thickness. These improvements typically yield more accurate CTF values that better represent the data. Although some of the improvements result in slower calculations per micrograph, these can be easily overcome through parallelization.There are some concerns outlined below that would benefit from further evaluation by the authors.For the examples shown in Figure 3b, given the small differences in estimated defocus1 and 2, what type of improvements would be expected in the reconstructed tomograms? Do such improvements in estimates manifest in better tilt-series reconstruction?

As explained in our preface, we do not believe that these difference would manifest in any improvements during tilt-series reconstruction and would not create any meaningful differences, even when tomograms are reconstructed with CTF correction. They might become meaningful during subtomogram averaging, but subtomograms are usually corrected using per-particle CTF estimation, similar to single-particle processing. We have included a new paragraph in the discussion to describe potential benefits of CTFFIND5 for cryo-tomography, Page 16 L390-399.

Similarly, the data shown in Figure 3C shows minimal improvements in the CTF resolution estimate (e.g., 4.3 versus 4.2 Å), but exhibited several hundred Å difference in defocus values. How do such differences impact downstream processing? Is such a difference overcame by per-particle (local) CTF refinements (like the authors mention in the discussion, see below)?

The difference in the defocus estimate (~600A) is substantially smaller than the thickness of the sample (2000A). Hence both estimates may be valid, depending on which particles inside the sample are considered. Particles with larger defocus errors could certainly be corrected by per-particle CTF refinement as long as the search range is chosen to be large enough. The main benefit of using CTFFIND5 is information for the user regarding the sample thickness to set the defocus search range appropriately.

At which point does the thickness of the specimen preclude the ice thickness modulation to be included for "accurate" estimate? 500Å? 1000Å? 2000Å? Based on the data shown in Figure 3B, as high as 969 Å thick specimens benefit moderately (4.6 versus 3.4 Å fit estimate), but perhaps not significantly, from the ice thickness estimation. Considering the increased computational time for ice thickness estimation, such an estimate of when to incorporate for single-particle workflows would be beneficial.

As explained in our preface, the main benefit for single-particle workflows will be sample tilt estimation. This will provide more accurate per-particle defocus estimates, compared to estimates that do not take the tilt into account. For single-particle samples, the ice thickness in holes is probably more efficiently monitored using the Beer-Lambert law.

It would seem that this statement could be evaluated herein: "the analysis of images of purified samples recorded at lower acceleration voltages, e.g., 100 keV (McMullan et al., 2023), may also benefit since thickness-dependent CTF modulations will appear at lower resolution with longer electron wavelengths". There are numerous examples of 300kV, 200kV, and 100kV EMPIAR datasets to be compared and recommendations would be welcomed.

Publicly available datasets recorded at 100kV and 200kV were collected in very thin ice, making it difficult to demonstrate the stated benefits. We have removed this statement.

Although logical, this statement is not supported by the data presented in this manuscript: "The improvements of CTFFIND5 will provide better starting values for this refinement, yielding better overall CTF estimation and recovery of high-resolution information during 3D reconstruction."

We have revised this statement and now explain that the sample tilt information will provide more accurate per-particle defocus estimates, compared to estimates that do not take the tilt into account, Page 17, L400-409. We did not investigate how this will affect downstream processing results.

Moreso, the lack of single-particle data evaluation does present a concern. Naively, these improvements would benefit all cryoEM data, regardless of modality.

We agree with the reviewer that all cryoEM modalities should benefit from more accurate defocus value estimates and have amended our concluding statement. However, how improved defocus values will benefit downstream processing results will depend on the processing pipeline, which includes various points of user input and data-dependent choices. We have therefore limited our analysis to the outputs of CTFFIND5.

**Recommendations for the authors:**

**Reviewer #1 (Recommendations For The Authors):**
(1) CTFFIND5 in cryo-ET(1.1) CTFFIND4 is prone to unreliable CTF estimates at high tilts in cryoET, a situation that can be identified by high variability or 'unstable' estimates as a function of the tilt angle. Prof. Mastronarde recently illustrated this situation in his article JSB 216:108057, 2024 (Fig. 7). Therefore, the authors could add results to show whether the improvements to tilt estimation introduced in CTFFIND5 overcome this problem. So, in addition to the estimation of tilt angle and tilt axis in Figure 2, the estimated defocus could also be shown.

We have worked with Prof. Mastronarde to help him use CTFFIND as a tool in his cryoET processing pipeline. Mastronarde chose CTFFIND because it contains algorithms and architecture that he could optimize for his purposes. CTFFIND5 is currently lacking the concept of a tilt series and can therefore not take advantage of the additional information that comes with tilt series. Our own applications for CTFFIND5 currently do not include tomography, and our results presented in Fig. 2 were obtained for validation of the tilt estimation feature. We did not attempt to duplicate Mastronarde’s optimization for reliable tilt series processing.

Figure 2b of this manuscript already suggests that CTFFIND5 may exhibit some variability of defocus estimates at high tilts (in view of the variability of tilt axis angle). A strategy used in IMOD and TOMOCTF is to consider the tiles of a group of consecutive images (typically 35; especially at high tilts) to add more signal to the average spectrum, thus providing more reliable estimates (illustrated in Mastronarde's article JSB 216:108057, 2024, Fig. 8). Will the authors think that CTFFIND5 might include a strategy like this for cryoET tilt-series?

We currently do not have plans to develop CTFFIND5 as a tool for tomography as there are already other excellent tools available, some of them based on CTFFIND’s basic algorithms (see previous comment).

(1.2) In cryoET, the CTF is often determined on the aligned tilt-series, with the tilt axis typically running along the Y axis. Has CTFFIND5 got the option to exclude estimation of the tilt geometry (tilt angle and/or axis) and, instead, take tilt geometry directly from the alignment and/or from the microscope??. This would significantly speed up determination of the CTF (in 1-2 seconds per image, according to Table 2) while still taking advantage of all power spectra in tilted images (as described in their tilt estimation algorithm) for improved CTF estimation. This strategy would be similar to what it is done in Bsoft and IMOD.

This is an excellent idea and we may implement this in an updated version. The current version is primarily meant for lamellae and single-particle samples where we usually have a single tilt in an unknown direction. For these cases, the suggested feature will have less benefit.

Thus, I suggest that the authors should also include results comparing CTF estimation in aligned tilt-series with CTFFIND4 and with CTFFIND5 (with no tilt estimation but indeed taking the tilt information from the alignment or the microscope into account). The results would show that CTFFIND5 is more robust than CTFFIND4, especially at high tilts.

Thank you for this suggestion. We are now showing a comparison of defocus estimates from CTFFIND4 and CTFFIND5 in Fig. 2. Indeed, in one case CTFFIND5 seems to report more robust defocus values at high tilt.

(1.3) The newer improvements in CTFFIND5 seem to be especially tailored to cryoET. The cryoET community will be highly attracted by these improvements. However, the current standard acquisition protocols (exposure of 3-5 e/A2 per image, tilts up to 60 degrees, etc) limit their full exploitation, particularly the thickness-aware CTF determination. I believe that adding a paragraph exclusively focused on cryoET and describing the potential benefits from CTFFIND5 and their limitations could enrich the Conclusion section. In this paragraph, the authors could highlight the great benefits from the tilt-aware CTF estimation. They could also discuss the current standard acquisition protocols (e.g. exposure 3-5 e/A2 per image, nominal defocus 3-5 microns, cellular thickness from 150 nm up to 200-300 nm that, at a tilt of 60 degrees, become 300 nm up to 400-600 nm) and their implications for the potential benefit from the improvements available in CTFFIND5.

This reviewer is clearly excited about the potential application of CTFFIND5 in cryoET. We are sorry that we are currently not developing CTFFIND5 in this direction.

(1.4) Apologies for insisting on cryoET in the previous points. I am just trying to suggest ideas to make CTFFIND5 even more helpful in cryoET. You can consider them now, or for a future version of the software, or just ignore them.

Thanks for your suggestions. Since there is clearly demand for tools to process tomographic tilt series, we will keep these suggestions in mind for the future development of CTFFIND.

(2) Tilt estimation(2.1) Page 4. Tiles for the initial steps in tilt estimation are of size 128x128. At which point tiles of larger size (e.g. 512x512) are used?. Please, define.

Thank you for pointing out this lack of clarity. For the tilt estimation, we used a tile size 128 x 128, which has been hard-coded in our program, as mentioned in line 68 on page4. For generating the final power spectrum, we usually use size 512 x 512. This tile size can be defined by the user when running the program. We have now clarified this on Page 4, L74-76.

(2.2) Page 6 and/or page 11: evaluation of tilt estimation with tilt-series.Please indicate the acquisition details of the tilt-series used for the evaluation, especially the exposure per image. This information is neither available in this manuscript nor in Elferich et al., 2022.Please, add these acquisition details similarly to page 9 in this manuscript (evaluation of sample thickness estimation using tomography): pixel size, exposure per image and total exposure, number of images, tilt range and interval

The same tilt-series were used to verify tilt-estimation and sample thickness. We have revised the Methods section to make this clear on Page5, L98-105 and Page 10, L202.

(2.3) Page 10. Section Results. Subsection Tilt estimation.The authors use "defocus correction" to refer to their method for scaling the power spectra. "Defocus correction" might perhaps be a misleading term. In contrast, in page 4 the authors use the term "tilt correction". Please, revise and make it consistent throughout the manuscript.

We agree and now use 'tilt correction' throughout the manuscript.

(2.4) Legend of Figure 2.Please add what the red dashed curve represents. Also, please note there might be an error in the estimated stage tilt axis angle: the legend states "171.8" where in the main text it is "178.2" (apparently, the latter is the correct one).

Thank you for pointing this out. We have modified the legend and changed the number in the legend to 178.2°.

(3) Thickness estimation(3.1) Line 141, page 7. The sentence reads: "The modulation of the CTF due to sample thickness t is described by the function E (current Equation 6), " I believe that the modulation envelope of the CTF due to sample thickness is not really E (current Equation 6), but the function sinc(E). Please, revise.

We have revised the manuscript as advised, Page 7, L148.

(3.2) Line 148, page 7. The sentence reads "an estimate of the frequency g of the first node of the CTF_t function "The concept of 'node' was introduced by Tichelaar et al. (2020). The authors should not assume that this concept is familiar to the readership. So, it is suggested that the authors should introduce this concept in this section. For instance, just after Equation 6 they could add a sentence like this: "This sinc modulation envelope increasingly attenuates the amplitude of the Thon rings with increasing spatial frequencies in an oscillatory fashion, with locations where the amplitude is zero known as nodes (Tichelaar et al., 2020)."

Thank you for this suggestion. We have revised the manuscript accordingly (Page 7, L151-156) and also marked the position of the first node in Fig. 3a.

(3.3) Line 154, page 8: A citation is lacking: "(corrected for astigmatism, as described in)". Perhaps the authors refer to the EPA (EquiPhase Averaging) method introduced by Zhang, JSB 193:1-12, 2016, 10.1016/j.jsb.2015.11.003.

Thanks for spotting this omission. We have added the appropriate reference.

(3.4) Figure 3.(3.4.1) Perhaps, the EPA (EquiPhase Averaging) method is used to reduce the 2D CTF to 1D curves, as represented in Figure 3b and 3c. Please, mention this in the legend of the figure or in the main text referring to Figure 3. The same might apply to Figure 1c.

Thanks for spotting this omission. We have clarified that this is indeed an EPA in the figure legends.

(3.4.2) Please indicate what the colored curves represent in 3b and 3c: The fitted CTF model (dashed red) and the EPA or astimatism-corrected radial average of power spectrum (solid black) ?

Thanks for spotting this omission. We have added descriptions of the colored lines in these plots (red = modeled CTF, blue = goodness of fit).

(3.4.3) Please note that the power spectrum (solid black curves in Figure 3b and 3c) does not look the same in the top and bottom panels: Without thickness estimation (top panels), the power spectrum is in the range [0,1] in Y, as expected. However, with thickness estimation (bottom panels), the power spectrum seems to have undergone a frequencydependent transformation (a rescaling or something that makes the power spectrum oscillates around 0.5 in Y). This transformation of the power spectrum resembles the thickness-induced sinc modulation of the CTF and seems to be appropriate to better fit the new thickness-aware CTF_t model in CTFFIND5 to the (transformed) power spectrum. However, this transformation of the power spectrum is not mentioned in the manuscript at all. Instead, according to the main text (page 8), the fitting method is based on the crosscorrelation between the new CTF model and the power spectrum, so I was expecting to see the same power spectrum black curve in the top and bottom panels. Please, clarify.

Indeed, CTFFIND5 displays the power spectrum differently after thickness estimation. We have revised the methods to explain this (page8, L178-181). The reviewer is also correct that the 1D lines plots of the Thon ring patterns in Fig. 3b and 3c are not identical. These 1D plots are generated from the 2D plots according to the fitted CTF, which is needed to follow the astigmatic rings and avoid blurring of the oscillations in the radial average. This means that different CTF fits will also result in somewhat different 1D plots. However, these differences only affect the 1D EPA plots shown to the user. The actual fitting is performed against the same 2D spectra.

(3.4.4) Line 319, Page 14. "A linear fit revealed .." It would be good to add a line with the linear fit in Figure 5.

Agreed. The revised Fig. 5 now shows a line for the linear fit.

(3.5) New CTF ModelIt is not clear from the text if the new CTF_t model is used at all times in CTFFIND5 or only when the user requests thickness estimation. Related to this, if the user requests both tilt estimation and thickness estimation, how is the CTF estimation process carried out in CTFFIND5?: Tilt and thickness are estimated at the same time? or one after the other (i.e. first the tilt is estimated, then followed by thickness estimation)?. Please, clarify.

The new CTF_t model is only used when the user requests thickness estimation. When both tilt-estimation and thickness estimation are requested, the tilt is estimated first and the corrected power spectrum is then fitted using the CTF_t model. We have revised the Methods section to explain this better, Page 8, L158-159.

(4) Pages 14-15. Section "CTF estimation and correction assists "This section just shows that correction of a highly underfocused image for the CTF with phase flipping or a Wiener filter reduces the CTF-induced fringes. I do not really understand the inclusion of this section to the manuscript. There is no contribution related to CTFFIND5.

The ability to apply a CTF correction to the input image according to Tegunov & Cramer is a new feature of apply_ctf, a program included with cisTEM. We think that this section fits into the theme of CTFFIND5 because the correction adds valuable information about the samples, such as FIB-milled lamellae.

If the authors prefer to keep this section, then please take the following points into account:(4.1) Figure 6b: This is the only time that the term "EPA" (EquiPhase Averaging, I guess) is used in the manuscript. Please, spell it out somewhere in the manuscript, define what it means and add a proper citation, if convenient. This point is related to point 3.3 above.

We have added the appropriate reference and defined EPA in the methods section as indicated in the reply to point 3.3.

(4.2) Figure 6d. The contrast of this image is poor. Please, increase the contrast (to be similar to Figure 6c) so that the details can be better discerned. The image also shows a grainy texture, likely artefacts from the Wiener filter due to excessive amplification. Maybe the 'strength parameter' S of the deconvolution Wiener filter (Tegunov & Cramer, 2019) should be tuned down or the 'fall-off parameter' F tuned up to try to attenuate these artefacts.

Agreed. The revised figure shows panel d with increased contrast with the custom fall-off parameter set to 1.3 and the custom strength parameter set to 0.7.

(5) CTFFIND5 runtimesTable 2 shows that estimation of tilt increases the runtime up to 39 s in an image of 4070x2892 and to 208 s in one of 2880x2046. There is a significant difference between these two cases (39 s vs. 208 s) and the first image is much larger than the second. Why does CTFFIND5 on the smaller image take so long compared to the larger image?

During tilt estimation, the images are binned to a pixel size of 5 Å. This causes micrograph 1 to be substantially smaller (in pixels) than micrographs 2 and 3, resulting in the faster runtime.

(6) Conclusions(6.1) In the Conclusion section, the authors could elaborate a bit the insights about the sample quality provided by CTFFIND5. This is stated in the title of the manuscript, but it was hardly mentioned in the manuscript.

We have revised the conclusion to make this clearer (Page 16, L389-396). CTFFIND5 helps in estimating sample quality since (1) the sample thickness is an important determinant in the amount of high-resolution signal in a micrograph and (2) the estimated fit-resolution reflects more accurately the amount of signal present in a micrograph after tilt and sample thickness have been taken into account.

(6.2) The authors nicely identify and describe the applications where thickness-aware CTF determination will be valuable: in situ single particle analysis and in vitro single particle cryoEM of purified samples at low voltages. Perhaps, CTFFIND5 will also be of great interest for single particle cryoEM of thick specimens (e.g. capsid of large viruses with diameter in the range 120-200 nm such as PBCV-1 or HSV-1).

Agreed. We have added this case to our Conclusions. (Fig. 3d)

(7) Typographical errors:line 161, page 8. "1.5 time" should be "1.5 times"lines 185-191. All exposures are given in 'electrons/Angstrom', not in 'electrons/square Angstrom'line 206, page 10. With "slides" the authors seem to mean "slices"line 338, page 14: "describeD by Tegunov"line 349, page 15. "power spectra"lines 366 and 368, page 15: Note that Square Angstrom is written as "A2". Put "2" with superscript.

Thank you for pointing out these errors. They have been corrected.

(8) References:Reference: Lucas et al., eLife 10 e68946. Year is lacking. Add year: 2021.Reference: Yan et al. 2015 cited in line 169, page 8, does not appear in Bibliography. The authors may mean: Yan et al. 2015 JSB 192:287-296, 2015It would be good to cite Bsoft, as it has a procedure similar to tilt-corrected CTF estimation: Heymann, Protein Science, 2021,

Thank you for carefully checking the cited references. We have revised the manuscript as suggested.

**Reviewer #2 (Recommendations For The Authors):**
I have only minor suggestions for improvement below:L218: "these option"

Corrected

L243: "chevron-shape" -> V-shape would be more accessible language for non-native speakers.

Changed

L281: "Based on these results we conclude that CTFFIND5 will provide more accurate CTF parameters" -> Given that the maximum resolutions of the fits by the old model and the new model are nearly the same, how big would the actual advantage of the new model be for subsequent sub-tomogram averaging?

Please see our response above, Reviewer #3 (Public Review),

L376: The correct reference for RELION per-particle CTF estimation is Zivanov et al, (2018) [https://elifesciences.org/articles/42166]. Also, the cryoSPARC paper referenced does not describe per-particle CTF estimation and should thus be removed from this context.

Thanks for pointing out these mistakes, which we have now corrected. We have chosen to keep the citation for CryoSPARC to reference the general software, but have added Ziavanov et.al. 2020 as suggested by the CryoSPARC website.

**Reviewer #3 (Recommendations For The Authors):**
Minor:Figure 1A legend - authors mention boxes but only 1 box is shown.

Thank you for pointing this out. For visual clarity we decided to only show one box. We have corrected the legend.

Figure 1B - it would be nice if the boxes that contributed to the power spectra were mapped on Figure 1A

The shown power spectra are not actual data. Instead, we show power spectra with exaggerated defocus differences for visual clarity. We have revised the figure legends to make this clear.

The Y-axis legends in Figure 2 are not aligned vertically

Corrected

Figure 3A - CTFFIND4 is missing an "I"

Corrected

Figure 3 - Y-axis legends are not aligned vertically

Corrected

Page 16, line 376, Relion should be RELION

We have revised the manuscript as advised.

Typo in equation 5, sinc versus sin?

'sinc' is correct here, since this is a thickness-dependent modulation of the CTF.

Lambert-Beer's, Lambert-Beer are used variably but curious if Beer-Lambert should be used.

We have revised the manuscript as advised.